# Multidisciplinary care for opioid dose reduction in patients with chronic non-cancer pain: A systematic realist review

Abhimanyu Sud[1]*, Alana Armas[2], Heather Cunningham[3], Shawn Tracy[2], Kirk Foat[4☯], Navindra Persaud[5,6☯], Fardous Hosseiny[7☯¤], Sylvia Hyland[8☯], Leyna Lowe[7], Erin Zlahtic[9‡], Rhea Murti[10‡], Hannah Derue[11‡], Ilana Birnbaum[12‡], Katija Bonin[12‡], Ross Upshur[2,13], Michelle L. A. Nelson[2,14]

1 Department of Family and Community Medicine, University of Toronto, Toronto, Ontario, Canada,
2 Bridgepoint Collaboratory for Research and Innovation, Lunenfeld-Tanenbaum Research Institute, Sinai Health System, Toronto, Ontario, Canada, 3 Gerstein Science Information Centre, University of Toronto, Toronto, Ontario, Canada, 4 Independent Researcher, London, Ontario, Canada, 5 Department of Family and Community Medicine, St. Michael's Hospital, Toronto, Ontario, Canada, 6 MAP Centre for Urban Health Solutions, Keenan Research Centre, St. Michael's Hospital, Toronto, Ontario, Canada, 7 Canadian Mental Health Association National, Toronto, Ontario, Canada, 8 Institute for Safe Medication Practices Canada, Toronto, Ontario, Canada, 9 Kinesiology, Western University, London, Ontario, Canada, 10 Arts & Science, McMaster University, Hamilton, Ontario, Canada, 11 Psychology, University of Guelph-Humber, Toronto, Ontario, Canada, 12 Faculty of Medicine, University of Toronto, Toronto, Ontario, Canada, 13 Dalla Lana School of Public Health, University of Toronto, Toronto, Ontario, Canada, 14 Institute of Health Policy, Management and Evaluation, University of Toronto, Toronto, Ontario, Canada

☯ These authors contributed equally to this work.
¤ Current address: Centre of Excellence on Post-Traumatic Stress Disorder, Ottawa, Ontario, Canada
‡ These authors also contributed equally to this work.
* Abhimanyu.sud@utoronto.ca

**Data Availability Statement:** All relevant data are within the manuscript and its Supporting Information files.

## Abstract

### Context

Opioid related deaths are at epidemic levels in many developed nations globally. Concerns about the contribution of prescribed opioids, and particularly high-dose opioids, continue to mount as do initiatives to reduce prescribing. Evidence around opioid tapering, which can be challenging and potentially hazardous, is not well developed. A recent national guideline has recognized this and recommended referral to multidisciplinary care for challenging cases of opioid tapering. However, multidisciplinary care for opioid tapering is not well understood or defined.

### Objective

Identify the existing literature on any multidisciplinary care programs that evaluate impact on opioid use, synthesize how these programs work and clarify whom they benefit.

### Study design

Systematic rapid realist review.

**Funding:** RU, MN, AS, SH, HC, NP received the Operating Grant: Opioid Crisis Knowledge Synthesis (OCK-156776) from the Canadian Institutes of Health Research (https://cihr-irsc.gc.ca/e/193.html). The funders had no role in study design, data collection and analysis, decision to publish, or preparation of the manuscript.

**Competing interests:** The authors have declared that no competing interests exist.

### Dataset

Bibliographic databases (MEDLINE, EMBASE, CINAHL, PsycINFO, Cochrane Library), grey literature, reference hand search and formal expert consultation.

### Results

95 studies were identified. 75% of the programs were from the United States and the majority (n = 62) were published after 2000. A minority (n = 23) of programs reported on >12 month opioid use outcomes. There were three necessary but insufficient mechanisms common to all programs: pain relief, behavior change and active medication management. Programs that did not include a combination of all three mechanisms did not result in opioid dose reductions. A concerning 20–40% of subjects resumed opioid use within one year of program completion.

### Conclusions

Providing alternative analgesia is insufficient for reducing opioid doses. Even high quality primary care multidisciplinary care programs do not reduce prescribed opioid use unless there is active medication management accomplished by changing the primary opioid prescriber. Rates of return to use of opioids from these programs are very concerning in the current context of a highly potent and lethal street drug supply. This contextual factor may be powerful enough to undermine the modest benefits of opioid dose reduction via multidisciplinary care.

### Introduction

Unintentional drug overdose deaths are high and on the rise throughout the Global North, particularly the developed economies of North America, Australia and several European countries. The United States (US) saw a nearly 10% rise in drug overdose deaths between 2016 and 2017. Nearly 70% of these deaths were attributable to opioids [1], resulting in an opioid mortality rate of 14.7 per 100,000. The comparable rate in Canada from 2018 is 12.0 per 100,000 and similar rates have been noted in other countries such as Sweden [2]. One large socioeconomic effect of these epidemics is that life expectancies in the US and Canada have decreased over the last several years. Other comparable countries, such as the United Kingdom (UK), Australia, Sweden and Finland, have demonstrated similar rises in years of life lost due to drug overdoses, though the absolute numbers are not as high as in North America [3]. Although the recent increase in mortality rate in North America appears to be driven by illicit opioids such as heroin and fentanyl, prescribed opioids continue to cause significant mortality [4]. In 2016, one third of opioid deaths in Ontario, Canada were directly related to a prescribed opioid and an additional third to diverted prescribed opioids [5].

High dose opioids prescribed for the long-term management of chronic pain increase the risk of death. One study examining approximately 10,000 patients with chronic pain in a US health management organization demonstrated that, as compared to people on a morphine equivalent (ME) daily dose of 1–20, those on doses of 20–50 ME had a 1.44-fold increased risk of overdose, those on doses of 50–100 ME were at a 3.73-fold increased risk, and those on >100 ME were at 8.87-fold increased risk [6]. Hazards such as death and overdose, medical

complications such as sleep apnea and hypogonadism, and side effects such as constipation and nausea, all demonstrate dose-related effects. The rise in North American high-dose prescribing has been identified as a major contributor to the current crisis and other countries appear to be following a similar trajectory. The UK has seen a 127% rise in high-dose opioid prescribing between 2008-2017 [7].

Given the iatrogenic contribution to these epidemics, one response from the medical and scientific communities has been to develop guidance documents around opioid prescribing, aiming to improve prescribing appropriateness and reduce consequent harms. Nationally applicable guidelines have been developed in the United States and Canada [8, 9]. While both draw from similar bodies of evidence and have a focus on dose limits for new prescribing, the Canadian guidelines made explicit recommendations for tapering, or dose reduction and used the GRADE approach to categorize recommendations as strong ("the recommendation can be adopted as policy in most situations") or weak ("policymaking will require substantial debate and involvement of various stakeholders"). These guidelines provided a weak recommendation that prescribers consider tapering opioids for people who are on greater than 90 ME per day. These same guidelines however, included a strong recommendation that "patients using opioids and experiencing serious challenges in tapering" should be referred to formal multidisciplinary care (MDC).

An important challenge of this last recommendation is that MDC for opioid dose reduction is not well defined by the guideline. Without a clear definition of what constitutes MDC, it is difficult to interpret and operationalize this recommendation. A recent systematic review of strategies for opioid dose reduction [10] identified effects of MDC chronic pain programs on opioid doses of about 10,000 patients across 31 studies. The review indicated significant heterogeneity with respect to program components, personnel, philosophical approaches, duration, and settings. Given this heterogeneity and the inherent complexity of the programs, the methods of traditional systematic review did not permit analysis beyond narrative description. Consequently, a significant knowledge gap remains regarding how MDC can most effectively be deployed to address high-dose opioid prescribing use and the opioid epidemic more generally. This is particularly challenging given that access to MDC for chronic pain management is severely limited even in well-resourced health systems [11].

The primary objectives for this systematic rapid realist review were to ascertain: what constitutes MDC for opioid dose reduction, for whom has this mode of care been evaluated, how does MDC for opioid dose reduction work, and in which contexts is MDC for opioid dose reduction effective or not effective?

## Materials and methods

Realist synthesis has been proposed as a method well-suited for examining heterogeneous and complex interventions [12]. An adaptation of this is the rapid realist review (RRR) which aims to review evidence to provide relevant knowledge readily applicable by policy makers and other knowledge users in a time and context sensitive manner [13]. Methodologically, Saul et al. note that RRRs work backwards "from the desired outcome to 'families of interventions' (I) that can be implemented to produce those outcomes, supported by a theoretical understanding of the contexts (C) within, and mechanisms (M) by which such interventions operate." Thus, our primary aim in this review was to determine which multidisciplinary programs (I) associate with reduced opioid doses (O), and to interpret how these reductions are achieved (M) and in which health system and social contexts (C).

The review was undertaken by a core team of health service researchers supported by an information scientist, local reference panel, and a series of clinical and research experts from

the United States and Canada. Collectively, the core research team, information scientist, and local reference panel have expertise in knowledge synthesis, pain management, opioid prescribing and tapering, clinical practice guideline development, interprofessional/collaborative/multidisciplinary practice, knowledge translation, and policy advocacy.

The entire review process, including research question, search strategy, screening, data extraction, and synthesis was subject to iterative review by the core research team and the local reference panel. We consulted 10 experts from the US and Canada, including investigators from recent systematic reviews and research in this area; clinicians who developed and deliver multidisciplinary care programs; health professionals working in multidisciplinary care settings as well as in primary care including physicians, psychologists and pharmacists; and, a patient with lived experience with opioid tapering and multidisciplinary care. These consultations aided in in identifying relevant literature, prioritizing outcomes of interest, and interpreting contextual and mechanistic factors.

Theoretical understanding was generated via an iterative, discursive process including the researchers, local reference panel, and experts. We aimed to keep theoretical understanding grounded in realism, and particularly the realism espoused by Pawson and Tilley [14] which asserts that causal mechanisms are to be identified at the level of human reasoning [15]. We first examined the Canadian guideline, a recent systematic review on opioid dose reduction [10], and associated knowledge translation products (e.g. an opioid tapering practice tool [16]). The most evident understanding identified through this process was one of analgesic substitution. Namely that since opioids provided some quantum of pain relief, if this quantum could be substituted by some non-opioid therapy, this would lead to less of a need for opioids and thus opioid dose reduction.

## Search strategy

The electronic database search strategy was created with the assistance of an Information Scientist (HC) with expertise in systematic reviews using controlled vocabulary and keywords representing the concepts "opioids", "dose reduction", "pain", and "multidisciplinary care". No limits on date, language, age or study design were set, but a filter to remove animal studies was applied. The strategy was peer reviewed and validated against a core set of 31 studies from the previously described systematic review [10]. Individual searches were conducted in Ovid MEDLINE, PsychINFO, AMED, CINAHL Plus and Cochrane Library between May and June 2018 (Ovid MEDLINE search strategy included in S1 Table).

We also conducted targeted searches of Ovid EMBASE (conference proceedings and meeting abstracts), Cochrane CENTRAL Trials database as well as ClinicalTrials.gov (clinical trial data), Proquest Dissertations and Theses (Dissertations), and four null and negative results journals. We conducted a search of the grey literature (reports and information not published commercially) of several dozen organizational websites. This was guided by the approach outlined by the Canadian Agency for Drugs and Technology [17]. The literature and grey literature searches were supplemented by scanning the reference lists of the core set of articles and contacting experts in the field. These additional searches were completed during June 2018. Lastly, throughout the project, hand searches of the references in relevant reviews were conducted.

In line with the iterative development of realist analysis, informal searches for new relevant literature published after the search dates were conducted throughout 2018 and 2019 to ensure the review findings reflected current literature. No additional studies were identified that substantively changed the review findings. Given this and the redundancy identified within the collected studies, we were confident that we have achieved data saturation and there was no further need to conduct an updated systematic database search.

## Study selection

To refine our search strategy, five articles [18–22] from the previous systematic review [10] were randomly selected and reviewed by members of the research team. These were used to identify preliminary Context, Mechanism and Outcome (CMO) configurations. Through this process we identified patient behaviour change, rather than analgesic substitution, as a driving mechanism and also that MDC may be conceptualized more as a contextual than mechanistic factor. The team also identified three possible outcomes to consider for the review: (1) opioid dose reduction, (2) pain management and (3) improved function. These interpretations were reviewed with the local reference panel for feedback. These initial understandings were also presented to national experts and stakeholders from the Canadian National Pain Faculty and feedback was used to further refine the theoretical understanding and inform the data extraction phase.

Studies of human subjects with chronic pain on prescribed opioids were included. Programs had to include MDC and the MDC intervention had to have been evaluated. Based on our team's extensive clinical and research experience with interprofessional and multidisciplinary care, particularly in primary care settings [23, 24], we acknowledged at study onset, and during the search and study selection processes that there were not consistent definitions or search terms for these complex concepts. For the purposes of this review, we defined MDC as any program that co-administered a non-opioid intervention alongside opioid prescribing and that included an opioid prescriber and a minimum of one other healthcare professional, as per the description provided in the Canadian guideline [8]. This is similar to definitions used for other reviews [25] and is distinct from "multimodal care" which does not require the involvement of additional healthcare providers. This definition of MDC does not require active collaboration between professionals, what the International Association for the Study of Pain defines as "interdisciplinary care", but is inclusive of this concept [26]. We included evaluations of any design that included an opioid dose outcome. We excluded studies that focused exclusively on patients who had cancer, were in palliative care, or who had opioid use disorder but not chronic pain. Articles reported in languages other than English were excluded.

We used Covidence©, a systematic review management platform, for screening. Titles and abstracts were screened in duplicate until an inter-rater reliability of 0.85 was achieved, after which we moved to a single screener. Disagreements were discussed at length and then adjudicated by a member of the research team with clinical and subject matter expertise (AS).

Full text review was done entirely in duplicate and an inter-rater reliability of 0.931 was achieved. All uncertainties and disagreements were discussed at length and then adjudicated by a member of the research team with clinical and subject matter expertise (AS).

Once screening had been completed, the core research team and local reference panel met at length to discuss insights from the search process and the expert consultations. At this point, the theoretical understanding was expanded to include the larger context that MDC programs are situated, such as the socio-political context of the time and place the MDC programs were run. The team also proposed to examine MDC programs in two different ways: MDC providing the environment for change (Context) and MDC driving the change (Mechanism). Finally, the team opted to prioritize the outcome of opioid dose reduction, given the context of opioid-related harms and since pain and function outcomes from MDC had been reviewed elsewhere [27].

## Data extraction

An initial data extraction form was created in Excel, in order to pilot test the extraction criteria. Data from a subset of studies was pilot extracted by multiple reviewers (AA, EZ, RH, HD, IB, KB), These extractions were compared and revised against the extractions of a subject matter

expert (AS). Once agreement on extraction criteria was achieved, the form was replicated in Covidence and data extraction was conducted singly. Five categories of data were collected: study identification (funding source, country, setting, authors, year of study and year of publication); methods (study design, research question, primary objective); population (inclusion/ exclusion criteria, group differences, baseline characteristics), intervention (duration, type of setting, program details, types of healthcare providers, program theory, control conditions, justification for MDC strategy, opioid tapering process), and outcomes (types, timeframes, indicators of acceptability to users, discussions of program success or failure).

When data extraction was completed, the core research team and local reference panel met to discuss the preliminary findings and a selection of studies, which were used to develop a preliminary configuration of outcome, intervention, context, and mechanism. This configuration was tested against the entire set of studies and used to inform the configuration presented here.

## Results

A total of 14,584 records were identified and 2,833 duplicates were removed, leaving 11,751 studies for title and abstract screening. 621 studies underwent full text review. Of these, 473 studies were excluded at full text review and another 53 studies were excluded during data extraction. A total of 95 studies met the inclusion criteria (Fig 1).

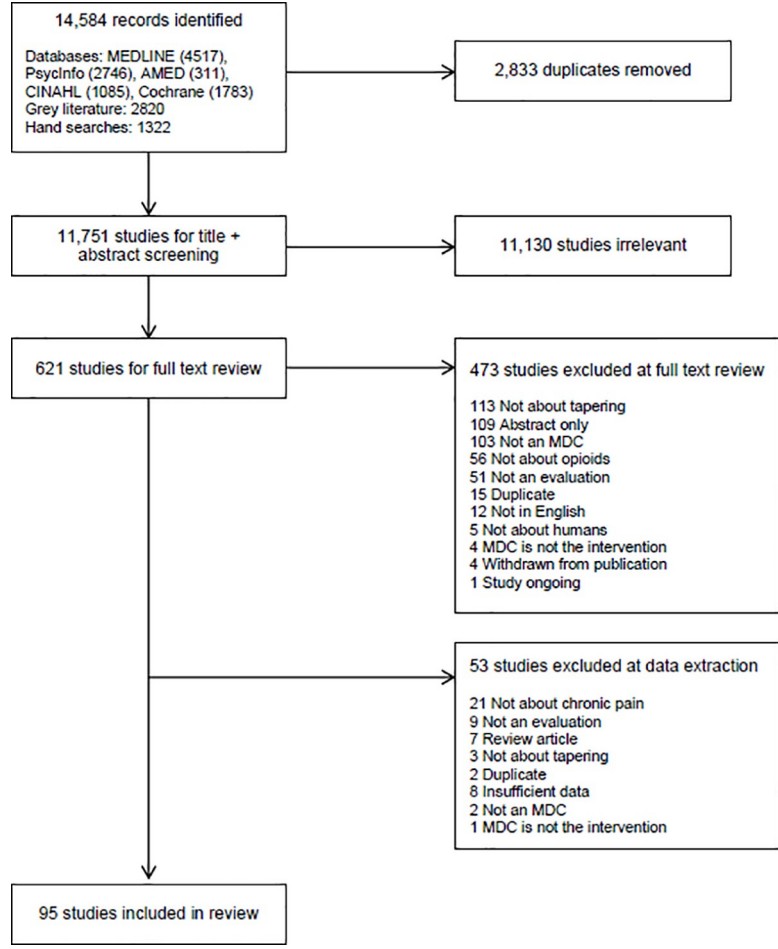

**Fig 1. PRISMA flow diagram.**

## Program characteristics

The included articles spanned nearly 50 years, with 43.6% published in the 2010s (Fig 2). The majority of studies took place in the United States (n = 71), see Table 1 for the other countries identified.

In the 95 studies, 96 evaluations were completed on 76 discrete MDC programs, as shown above. Fourteen evaluations took place at the Mayo Clinic's Pain Rehabilitation Centre [20, 43, 45, 46, 48, 59, 64, 65, 79, 108, 109] and seven other programs had two evaluations completed [32, 39–41, 56, 57, 63, 69, 70, 73, 74, 99, 100, 112]. 67.1% of programs were situated in tertiary care, academic settings. There were outpatient (43%), inpatient (30%), and mixed outpatient/ inpatient (15%) programs.

The most common study designs were prospective cohort studies (28.4%), retrospective cohort studies (28.4%), randomized control trials (16.8%), and case studies (12.6%).

Program duration was variable, with the shortest program running between 1 and 5 days [83], to the longest program of 14 months [42]. The modal program length was between 3 and 4 weeks (n = 22 programs) and program durations between 2–4 months were the second most common (n = 11 programs). There were a variety of follow-up periods used to examine the sustainability of immediate post-program outcomes post program. Of the 96 included evaluations, 43 measured outcomes only during the program or at program completion. Of the remaining evaluations (n = 53), outcomes were measured anywhere between one to three months post program [33, 35, 45, 62, 90, 96, 116] to over 5 years from program completion [61, 71] (Fig 3). Many of the evaluations measured outcomes at multiple follow up periods.

Thirty-three types of healthcare providers and staff were identified. Besides physicians, the most common were psychologists (n = 42), physiotherapists (n = 37) and nurses (n = 28). There was large variation in the size of the MDC teams. The largest MDC team consisted of 18 team members [38] and the smallest teams consisted of one healthcare provider in addition to the prescriber [36, 49, 60, 72, 74, 116]. The most common team sizes included 4 to 6 healthcare providers and staff (n = 21). Data about program staffing was missing for 19 programs (25%).

Forty-four (57.9%) programs had a required opioid tapering protocol, another 7 (9.2%) had suggested tapering protocols but did not require tapering (Fig 4). The remaining MDC

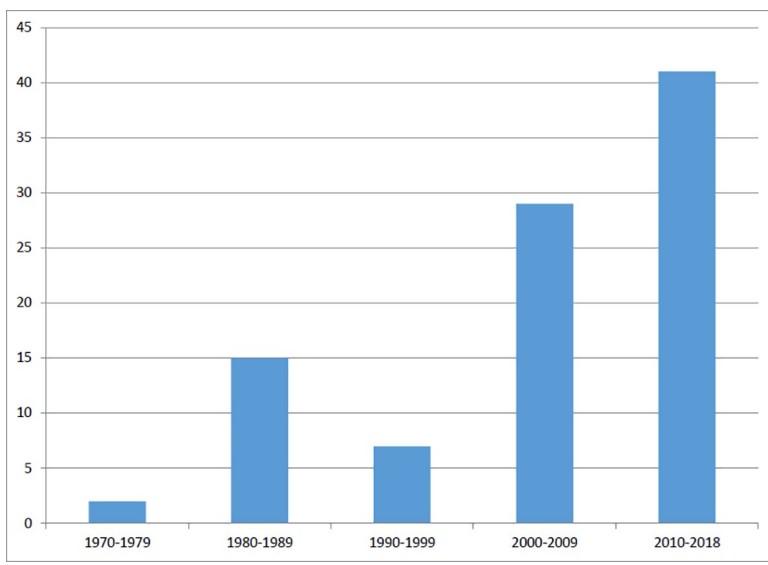

**Fig 2. Included studies by decade.**

**Table 1. Included studies characteristics.**

| Authors | Year | Country | Design | Program | Type of Institution | Care Level | Clinical Setting | Duration / Contact Time | Opioid Taper Protocol | Time of Last Outcome Measurement* |
|---|---|---|---|---|---|---|---|---|---|---|
| Angeles et al. [28] | 2013 | Canada | Randomized controlled trial | McMaster Family Health Team Clinics | Academic | Primary | Outpatient | 8 weeks / 16hrs | No protocol | 1–11 months |
| Atkins et al. [29] | 2014 | USA | Ad hoc study design | Rancho Los Amigos National Rehabilitation Center | Community | Tertiary | Inpatient | 3 weeks / Unclear | Suggested taper | Discharge |
| Barr [30] | 2016 | USA | Prospective cohort study | Inpatient psychiatric treatment facility | INP⁺ | Tertiary | Inpatient | 2–34 days / 8–136hrs | No protocol | Discharge |
| Bass et al. [31] | 2007 | UK | Retrospective cohort study | Regional pain clinic | Academic | Tertiary | Outpatient | Not specified / INP | Suggested taper | 12+ months |
| Becker et al. [32] | 2000 | Denmark | Randomized controlled trial | Copenhagen University Hospital Multidisciplinary Pain Centre | Community | Tertiary | Outpatient | 10.5 months / INP | No protocol | 1–11 months |
| Belkin et al. [33] | 2017 | USA | Retrospective cohort study | Multidisciplinary Pain Service, Addiction Medicine at SUNY Hospital, New York. | Academic | Primary | Outpatient | Unclear / Unclear | Required taper | 1–11 months |
| Bruce et al. [34] | 2009 | USA | Prospective cohort study, with a case example | Comprehensive Pain Rehabilitation Program, Mayo Clinic | Academic | Tertiary | Inpatient | 3 weeks / INP | Required taper | Discharge |
| Bruce et al. [35] | 2017 | USA | Controlled interrupted time series | Interdisciplinary Pediatric Pain Rehabilitation Program, Mayo Clinic | Academic | Tertiary | Outpatient | 3 weeks / 120hrs | Required taper | Discharge |
| Chandwani et al. [36] | 2008 | USA | Case study | Personalized pain program | Academic | Tertiary | Outpatient | 8 months / Unclear | Required taper | During treatment |
| Chapman et al. [37] | 1981 | USA | Randomized controlled trial | Pain Control Centre, Department of Rehabilitation Medicine, Emory University, Atlanta, GA | Academic | Tertiary | Outpatient | 2–6 weeks / 18hrs | Required taper | 12+ months |
| Cinciripini & Floreen [38] | 1982 | USA | Prospective cohort study | Miller-Dwan Pain Program | Academic | Tertiary | Inpatient | 4 weeks / 300hrs | Required taper | 12+ months |
| Clark et al.(a) [39] | 2009 | USA | Retrospective cohort study | Tampa Polytrauma Rehabilitation Center | Academic | Tertiary | Inpatient | Unclear / Unclear | Suggested taper | Discharge |
| Clark et al.(b) [40] | 2009 | USA | Case study | Tampa Polytrauma Rehabilitation Center | Academic | Tertiary | Inpatient | 30 days / Unclear | Suggested taper | 1–11 months |
| Clarke et al. [41] | 2018 | Canada | Single centre observational study | Transitional Pain Service | Academic | Tertiary | Outpatient | INP / INP | Required taper | 1–11 months |
| Cowan et al. [42] | 2003 | UK | Retrospective cohort study | Pain clinic, London district general hospital | Community | Tertiary | Inpatient & outpatient | 14 months / INP | Unclear | Discharge |
| Crisostomo et al. [43] | 2008 | USA | Retrospective cohort study | Comprehensive Pain Rehabilitation Program, Mayo Clinic | Academic | Tertiary | Outpatient | 3 weeks / 120hrs | Required taper | Discharge |
| Cucchiaro et al. [44] | 2013 | USA | Case study | Chronic Rehabilitation Unit, Children's Hospital Los Angeles | Community | Tertiary | Inpatient | 9 days / INP | Required taper | 12+ months |
| Cunningham et al. [45] | 2009 | USA | Retrospective cohort study | Comprehensive Pain Rehabilitation Program, Mayo Clinic | Academic | Tertiary | Outpatient | 3 weeks / INP | Required taper | Discharge |
| Cunningham et al. [46] | 2016 | USA | Retrospective cohort study | Comprehensive Pain Rehabilitation Program, Mayo Clinic | Academic | Tertiary | Outpatient | 3 weeks / 120hrs | Required taper | Discharge |

(*Continued*)

**Table 1.** (Continued)

| Authors | Year | Country | Design | Program | Type of Institution | Care Level | Clinical Setting | Duration / Contact Time | Opioid Taper Protocol | Time of Last Outcome Measurement* |
|---|---|---|---|---|---|---|---|---|---|---|
| Currie et al. [47] | 2003 | Canada | Randomized controlled trial | University of Calgary Addiction Centre Pain Management Program | Academic | Tertiary | Outpatient | 10 weeks / 15hrs | Required taper | 12+ months |
| Darchuk et al. [48] | 2010 | USA | Quasi-experimental time series | Comprehensive Pain Rehabilitation Program, Mayo Clinic | Academic | Tertiary | Outpatient | 3 weeks / INP | Required taper | 1–11 months |
| Davis et al. [49] | 2018 | USA | Prospective pragmatic intervention trial | Private offices of acupuncturists, Vermont | Community | Tertiary | Outpatient | 60 days / INP | No protocol | Discharge |
| De Williams et al. [50] | 1993 | UK | Prospective cohort study | Pain management unit, UK Hospital | Community | Tertiary | Inpatient & outpatient | 4 weeks / 170hrs | Required taper | 1–11 months |
| Deardorff et al. [51] | 1991 | USA | Case-control study | California Pain Center | Academic | Tertiary | Inpatient & outpatient | Variable / INP | Required taper | 1–11 months |
| Dersh et al. [52] | 2008 | USA | Prospective cohort study | Interdisciplinary functional restoration program, regional referral center | Academic | Tertiary | Outpatient | Unclear / INP | Required taper | During treatment |
| Dolce et al. [53] | 1986 | USA | Prospective cohort study | Multidisciplinary pain management program | Community | Tertiary | Inpatient & outpatient | 4 weeks / 160hrs | Required taper | 1–11 months |
| Doolin [54] | 2017 | USA | Prospective cohort study | Program in a California corrections facility | N/A | Tertiary | N/A | 60 days / INP | Suggested taper | Discharge |
| Eng & Lachenmeyer [55] | 1996 | USA | Case study | Personalized pain program | Community | Tertiary | INP | 8 months / INP | Required taper | During treatment |
| Finlayson et al. (a) [56] | 1986 | USA | Prospective cohort study | Mayo Clinic and an affiliated Alcohol and Drug Dependence Unit | Academic | Tertiary | Inpatient | 28 days / INP | Required taper | 12+ months |
| Finlayson et al. (b) [57] | 1986 | USA | Prospective cohort study | Mayo Clinic and an affiliated Alcohol and Drug Dependence Unit | INP | INP | INP | INP / INP | Not reported | 12+ months |
| Fordyce et al. [58] | 1973 | USA | Prospective cohort study | Hospital-based comprehensive medical rehabilitation center | Academic | Tertiary | Inpatient & outpatient | 3–7 weeks / INP | Required taper | Discharge |
| Gilliam et al. [59] | 2018 | USA | Randomized controlled trial | Comprehensive Pain Rehabilitation Program, Mayo Clinic | Academic | Tertiary | Outpatient | 3 weeks / 120hrs | Required taper | 1–11 months |
| Groessl et al. [60] | 2017 | USA | Randomized controlled trial | VA Medical Center, California | Academic | Tertiary | Outpatient | 12 weeks / 24hrs | No protocol | 1–11 months |
| Guck et al. [61] | 1985 | USA | Randomized controlled trial | Nebraska Pain Management Center | Academic | Tertiary | Inpatient | 4 weeks / INP | Required taper | 12+ months |
| Hassamal et al. [62] | 2016 | USA | Case study | Spine Centre | Academic | Tertiary | Outpatient | 6–8 weeks / INP | Required taper | 1–11 months |
| Hojsted et al. [63] | 2006 | Denmark | Prospective cohort study | Copenhagen University Hospital Multidisciplinary Pain Centre | Academic | Tertiary | INP | 3 months / INP | No protocol | Discharge |
| Hooten et al. (a) [64] | 2007 | USA | Prospective case series | Comprehensive Pain Rehabilitation Program, Mayo Clinic | Academic | Tertiary | Outpatient | 3 weeks / 120hrs | Required taper | Discharge |
| Hooten et al. (b) [65] | 2007 | USA | Retrospective case-matched series | Comprehensive Pain Rehabilitation Program, Mayo Clinic | Academic | Tertiary | Outpatient | 3 weeks / 120hrs | Required taper | Discharge |

(*Continued*)

**Table 1.** (Continued)

| Authors | Year | Country | Design | Program | Type of Institution | Care Level | Clinical Setting | Duration / Contact Time | Opioid Taper Protocol | Time of Last Outcome Measurement* |
|---|---|---|---|---|---|---|---|---|---|---|
| Hooten et al. [66] | 2009 | USA | Retrospective cohort study | Comprehensive Pain Rehabilitation Program, Mayo Clinic | Academic | Tertiary | Outpatient | 3 weeks / 120hrs | Required taper | Discharge |
| Hooten et al. [67] | 2010 | USA | Prospective cohort study | Comprehensive Pain Rehabilitation Program, Mayo Clinic | Academic | Tertiary | Outpatient | 3 weeks / 120hrs | Required taper | Discharge |
| Hubbard et al. [68] | 1996 | USA | Prospective cohort study | Pain management program in an outpatient multidisciplinary component of a large neurology private practice | Academic | Tertiary | Outpatient | 2–4 weeks / INP | Not reported | Discharge |
| Huffman et al. [69] | 2013 | USA | Retrospective cohort study | Cleveland Clinic | Academic | Tertiary | Outpatient | 3–4 weeks / INP | Required taper | 12+ months |
| Huffman et al. [70] | 2017 | USA | Retrospective cohort study | Cleveland Clinic | Academic | Tertiary | Outpatient | 3–4 weeks / 142.5–190hrs | Required taper | 12+ months |
| Jensen et al. [71] | 2005 | Denmark | Retrospective cohort study | Danish multidisciplinary pain centre | Academic | Tertiary | Outpatient | 8 months / INP | Not reported | 12+ months |
| Keefe et al. [72] | 1981 | USA | Retrospective cohort study | Behavioral Physiology Laboratory, Duke University Medical Center | Academic | Tertiary | INP | INP / INP | Suggested taper | Discharge |
| Khatami et al. [73] | 1979 | USA | Randomized controlled trial | Multimodal treatment for chronic pain | Academic | Tertiary | Outpatient | 31.2 weeks / 31hrs | No protocol | Discharge |
| Khatami & Rush [74] | 1982 | USA | Case-control study | Multimodal treatment for chronic pain | Academic | Tertiary | Outpatient | 6.5–23 weeks / 7–23hrs | No protocol | 1–11 months |
| Kidner et al. [75] | 2009 | USA | Prospective cohort study | Interdisciplinary functional restoration program, Texas | Academic | Tertiary | INP | INP / INP | Required taper | Discharge |
| Kroening & Oleson [76] | 1985 | USA | Systematic case study | UCLA Pain Management Center | Academic | Tertiary | Inpatient & outpatient | Unclear / Unclear | Required taper | 12+ months |
| Kroenke et al. [77] | 2009 | USA | Randomized controlled trial | SCAMP | Academic | Tertiary | Outpatient | 12 months / INP | No protocol | 12+ months |
| Krumova et al. [78] | 2013 | Germany | Retrospective cohort study | University Hospital Bergmannsheil Department for Pain Management | Academic | Primary | Inpatient | 22 days / INP | Required taper | 12+ months |
| Kurklinsky et al. [79] | 2016 | USA | Retrospective cohort study | Comprehensive Pain Rehabilitation Program, Mayo Clinic | Academic | Tertiary | Outpatient | 3 weeks / Unclear | Required taper | Discharge |
| Lake 3rd et al. [80] | 2009 | USA | Retrospective cohort study | Chelsea Community Hospital | Academic | Tertiary | Inpatient & outpatient | 13 days / INP | Required taper | Discharge |
| Levendusky & Pankratz [81] | 1975 | USA | Case study | Veterans Administration Hospital, Portland, Oregon | Academic | Primary | Inpatient | INP / INP | Required taper | 1–11 months |
| Linton & Melin [82] | 1983 | Sweden | Randomized controlled trial | Physical Rehabilitation ward at a major Swedish hospital | Academic | Primary | Outpatient | INP / INP | No protocol | Discharge |
| Maani et al. [83] | 2011 | USA | Retrospective cohort study | Army Burn Center, Brooke Army Medical Center | Academic | Tertiary | Inpatient | 1–5 days / INP | Required taper | Discharge |

(*Continued*)

**Table 1.** (Continued)

| Authors | Year | Country | Design | Program | Type of Institution | Care Level | Clinical Setting | Duration / Contact Time | Opioid Taper Protocol | Time of Last Outcome Measurement* |
|---|---|---|---|---|---|---|---|---|---|---|
| MacLaren et al. [84] | 2005 | USA | Prospective cohort study | Interdisciplinary functional restoration program—West Virginia | Academic | Tertiary | INP | 4–6 weeks / 120–180hrs | No protocol | 1–11 months |
| Malec et al. [85] | 1981 | USA | Retrospective cohort study | Inpatient pain management program for chronic benign pain | INP | Tertiary | Inpatient | INP / INP | Required taper | 12+ months |
| Meana et al. [86] | 1999 | USA | Case study | Personalized pain program | Academic | Primary | INP | 12 weeks / INP | Required taper | Discharge |
| Mehl-Madrona et al. [87] | 2016 | USA | Case-control study | Group medical visits program, primary care clinic | Academic | Primary | Outpatient | 7.4 months / INP | Required taper | Discharge |
| Mudge et al. [88] | 2016 | Australia | Prospective cohort study | THRIVE Program | Academic | Primary | Outpatient | 12 weeks / INP | Suggested taper | Discharge |
| Murphy et al. [89] | 2013 | USA | Retrospective cohort study | Veteran Affairs hospital | Academic | Primary | Inpatient | 3 weeks / 90hrs | Required taper | Discharge |
| Murphy et al. [90] | 2016 | USA | Retrospective cohort study | Chronic Pain Rehabilitation Program, James A. Haley Veterans' Hospital | Academic | Tertiary | Inpatient | 3 weeks / 90–120hrs | Required taper | 1–11 months |
| Nissen et al. [18] | 2001 | Australia | Retrospective cohort study | Royal Brisbane Hospital Multidisciplinary Pain Centre | Academic | Tertiary | Inpatient | 2 weeks / INP | Required taper | Discharge |
| Oohata et al. [91] | 2017 | Japan | Retrospective cohort study | Outpatient pain clinic | Academic | Tertiary | Outpatient | Variable / INP | No protocol | Discharge |
| Philips [92] | 1987 | Canada | Randomized controlled trial | Behavioral treatment program | Academic | Tertiary | Outpatient | 9 weeks / INP | Not reported | 12+ months |
| Portnow et al. [93] | 1985 | USA | Case study | Medically Induced Drug Addiction Center connected with a comprehensive Pain Center at a major teaching hospital | Academic | Tertiary | Outpatient | INP / INP | Required taper | 1–11 months |
| Ralphs et al. [19] | 1994 | UK | Randomized controlled trial | Pain clinic, St. Thomas Hospital | Academic | Tertiary | Inpatient | 4 weeks / INP | Required taper | 1–11 months |
| Rome et al. [20] | 2004 | USA | Retrospective cohort study | Comprehensive Pain Rehabilitation Program, Mayo Clinic | Academic | Tertiary | Outpatient | 3 week / INP | Required taper | Discharge |
| Ruhe et al. [94] | 2017 | Germany | Retrospective cohort study | German paediatric pain centre | Academic | Tertiary | Inpatient | 3 weeks / INP | Unclear | 12+ months |
| Seal et al. [95] | 2017 | USA | Matched case control | Integrated Pain Team in primary care, San Francisco VA Health Care System | Academic | Primary | INP | INP / INP | Not reported | 1–11 months |
| Seres & Newman [96] | 1976 | USA | Prospective cohort study | Portland Pain Center | INP | INP | Inpatient | 15–25 days / INP | Not reported | 1–11 months |
| Sime [97] | 2004 | USA | Case study | Personalized pain program | Community | Tertiary | Outpatient | 9 months / INP | No protocol | 12+ months |
| Smith et al. [98] | 1988 | USA | Prospective cohort study | Emanuel Pain Centre | Academic | Tertiary | INP | 3 weeks / INP | Not reported | 12+ months |
| Snow et al. [99] | 1986 | USA | Prospective cohort study | Hospital for Joint Diseases Orthopedic Institute's Orthopedic Arthritis Pain Center | Academic | Tertiary | Inpatient | 3 weeks / INP | Not reported | 12+ months |

(*Continued*)

**Table 1.** (Continued)

| Authors | Year | Country | Design | Program | Type of Institution | Care Level | Clinical Setting | Duration / Contact Time | Opioid Taper Protocol | Time of Last Outcome Measurement* |
|---|---|---|---|---|---|---|---|---|---|---|
| Snow et al. [100] | 1988 | USA | Retrospective cohort study | Hospital for Joint Diseases Orthopedic Institute's Orthopedic Arthritis Pain Center | Academic | Tertiary | Inpatient & outpatient | 3 weeks / INP | Unclear | 12+ months |
| Sundaraj et al. [101] | 2005 | Australia | Retrospective cohort study | Pain Management Centre, Nepean Teaching Hospital | Academic | Tertiary | Outpatient | INP / INP | No protocol | 12+ months |
| Taylor et al. [102] | 1980 | USA | Prospective cohort study | University of Utah Pain Clinic | Academic | Primary | Inpatient | 11 days / Unclear | Required taper | 1–11 months |
| Tennant Jr & Rawson^ [103] | 1982 | USA | Prospective cohort study | 1. Detoxification and counselling outpatient treatment program | Academic | Tertiary | Outpatient | 3 weeks / INP | Required taper | 1–11 months |
| | | | | 2. Detoxification and maintenance outpatient treatment program | Academic | Tertiary | Outpatient | 3–18 months / Unclear | Required taper | 1–11 months |
| Thieme et al. [21] | 2003 | Germany | Randomized controlled trial | Inpatient program at a hospital for rheumatic disorders | Academic | Tertiary | Inpatient | 5 weeks / 75hrs | Required taper | 12+ months |
| Thorn et al. [104] | 2007 | USA | Prospective cohort study | Kilgo Headache Clinic | Academic | Tertiary | Outpatient | 10 weeks / 15hrs | No protocol | Discharge |
| Tiipana et al. [105] | 2016 | Finland | Retrospective cohort study | The Acute Pain Service Out-Patient Clinic | Academic | Tertiary | Outpatient | 2.8 months / INP | Required taper | Discharge |
| Timmings et al. [106] | 1980 | USA | Prospective cohort study | Pain Management Program in the Rehabilitation Medicine Program | Academic | Tertiary | Inpatient | 4–6 weeks / Unclear | Required taper | Discharge |
| Tollison et al. [107] | 1985 | USA | Clinical outcome investigation | Pain Therapy Center, Greenville Hospital System, South Carolina | Community | Tertiary | Inpatient | 21–28 days / INP | Suggested taper | 12+ months |
| Townsend et al. [108] | 2008 | USA | Prospective cohort study | Comprehensive Pain Rehabilitation Program, Mayo Clinic | Academic | Tertiary | Outpatient | 3 weeks / INP | Suggested taper | Discharge |
| Townsend et al. [109] | 2006 | USA | Case study | Comprehensive Pain Rehabilitation Program, Mayo Clinic | Academic | Tertiary | Outpatient | 3 weeks / INP | Required taper | Discharge |
| Tyre & Anderson [110] | 1981 | USA | Prospective cohort study | Pain Management Service, Waukesha Hospital | Community | Tertiary | Inpatient | 3–6 weeks / Unclear | Required taper | Discharge |
| Vines et al. [111] | 1996 | USA | Prospective cohort study | Outpatient Chronic Pain Program | Academic | Tertiary | Inpatient & outpatient | 4 weeks / 160hrs | Unclear | 1–11 months |
| Weinrib et al. [112] | 2017 | Canada | Case study | Transitional Pain Service | Academic | Tertiary | Inpatient & outpatient | 3–6 months / INP | Required taper | 1–11 months |
| Williams et al. [22] | 1996 | UK | Randomized controlled trial | Pain management unit in the UK | Academic | Tertiary | Inpatient & outpatient | 4 weeks / INP | Required taper | 12+ months |
| Worzer [113] | 2015 | USA | Retrospective cohort study | Chronic pain program | Academic | Tertiary | INP | INP / INP | Required taper | Discharge |
| Younger et al. [114] | 2008 | USA | Prospective cohort study | Stanford Comprehensive Interdisciplinary Pain Program | Academic | Tertiary | Inpatient | 7–14 days / INP | Required taper | Discharge |
| Zheng et al. [115] | 2008 | Australia | Randomized controlled trial | Pain Management Centre, St. Vincent Hospital | Academic | Tertiary | Outpatient | 6 weeks / 6hrs | No protocol | 1–11 months |

*(Continued)*

**Table 1.** (Continued)

| Authors | Year | Country | Design | Program | Type of Institution | Care Level | Clinical Setting | Duration / Contact Time | Opioid Taper Protocol | Time of Last Outcome Measurement* |
|---|---|---|---|---|---|---|---|---|---|---|
| Zheng et al. [116] | 2018 | Australia | Randomized controlled trial | Pain program run through Pain Services Unit, Melbourne Hospital and 4 other sites in Victoria | Academic | Tertiary | Outpatient | 10 weeks / 4hrs | Not reported | 1–11 months |
| Zhou et al. [117] | 2017 | USA | Case study | Multidisciplinary pain treatment | Academic | Tertiary | Outpatient | INP / INP | Required taper | 1–11 months |

*During = measured during treatment, Discharge = measured at program completion, 1–11 months and 12+ months = post-program.

⁺INP = Information not provided.

^Tennant and Rawson [103] presented evaluations on two different programs in their study.

programs did not have a tapering protocol (18.4%), did not report on a tapering protocol (9.2%), or were unclear about their protocol (5.3%). The large amount of variation in tapering protocols precluded any meaningful groupings. 58.3% of evaluations used harms from opioids as a primary rationale for the program and its evaluation.

The prototypical program was an outpatient, full-time, 3 to 4-week multimodal chronic pain program based at an academic, tertiary care centre that required opioid tapering as an essential, and usually preliminary, part of the program. The prototypical program included at least a psychologist, physical therapist and specialist physician working as a coordinated team with no specific coordination back to the referring provider. Evaluations of these programs were most concerned with outcomes at program completion (i.e. at 3–4 weeks).

## Synthesis–what works for whom in what contexts?

The following section presents a synthesis of the findings into eight statements that resulted from the development of the CMO configuration for this review. Table 2 provides a summary of these statements.

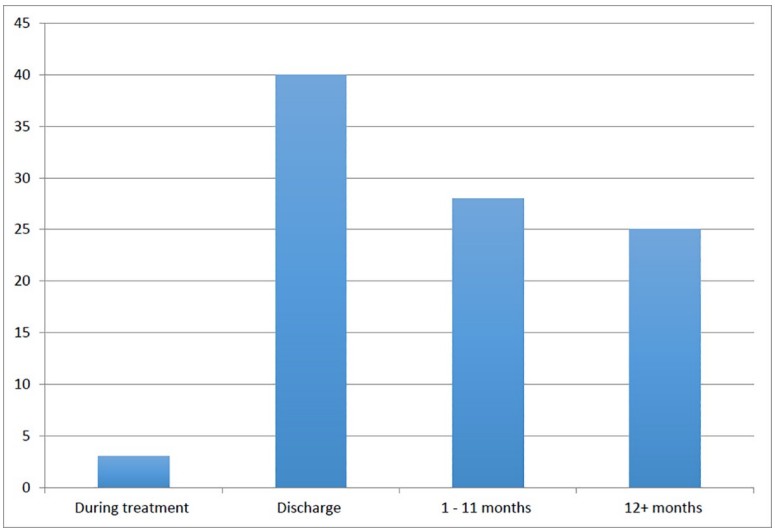

**Fig 3. Time of last outcome measurement.**

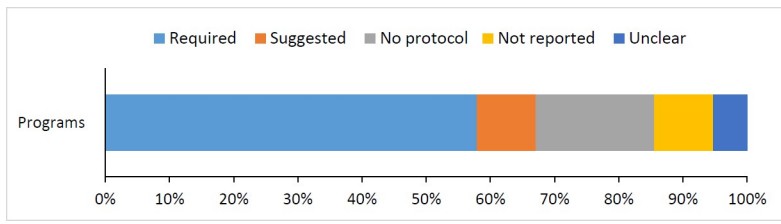

**Fig 4. Programs' opioid tapering protocols.**

**Participant characteristics.**   The majority of programs (60.5%) included patients with all chronic pain types, who had no evidence of progressive disease or active underlying disease, who had greater than six months of pain of various severities, and whom typically had a long history of engagement with the healthcare system, including previous surgeries for their pain [61, 99, 100]. An exemplar study reported that patients fell into three groups of pain types and locations (back pain, head pain and other) and a mean chronicity of pain of 8.6 years (SD 7.8) [92]. While most studies examined patients who had persistent pain despite continuous opioid use, some [47, 52, 56, 57, 69, 73, 76, 93, 103, 112] attempted to examine specifically patients who had evidence of opioid use disorder in the context of chronic pain. Other terms used in the studies included opioid dependence, drug dependence/addiction, and narcotic addiction, but for the purposes of this paper we grouped these under opioid use disorder.

## Synthesis statement 1: MDC for opioid dose reduction has been evaluated primarily for people with persistent chronic pain who have had long-term engagement with the health care system for management of their pain

**Program design and justification: What works?.**   We identified three intervention components that consistently patterned with successful programs as well as counterfactual cases: 1)

**Table 2. Synthesis statements.**

| | Synthesis Statement |
|---|---|
| 1. | MDC for opioid dose reduction has been evaluated primarily for people with persistent chronic pain who have had long-term engagement with the health care system for management of their pain. |
| 2. | Pain management using various modalities such as physical therapy, nerve blocks, and acupuncture is an essential component to MDC programs that reduce opioid doses. It was unclear which modality had the greatest opioid reducing effect in terms of number of respondents, total dose reductions or long-term abstinence. |
| 3. | Patient-focused behaviour change is a very common intervention and frequently a guiding principle for MDC programs that result in opioid dose reduction. |
| 4. | Programs that do not directly manipulate opioid doses are unlikely to reduce opioid use. A greater focus on opioid dose reduction was associated with larger opioid dose reductions. National and health system attitudes towards opioids directly influence the presence and intensity of opioid tapering components. |
| 5. | MDC for opioid dose reduction has been studied with people who are willing to actively engage in tapering of opioids. |
| 6. | Treatment setting (tertiary versus primary care) is not determinative of outcomes. Programs that did not change the prescriber, regardless of treatment setting, were unlikely to be successful in reducing opioid doses. |
| 7. | Chronic pain MDC programs provide an amenable context for analgesia, patient behaviour change, and opioid tapering to co-occur. Given the contact time required for such programs, they are more likely to occur in resource intensive settings, such as tertiary care centres. |
| 8. | Return to opioid use after complete discontinuation was common. Factors that have been investigated as influencing the likelihood of return to use included pain levels, depression severity, functionality, and degree of self-efficacy. |

pain relief via physical interventions, 2) patient behaviour modification, and 3) changing the opioid prescribing pattern via changing the prescriber.

*Pain relief is a motivating force for program development and patient participation.* All included programs were focused on providing pain relief and 66 studies specifically reported pain-related outcomes. They used a variety of analgesic approaches, including rehabilitative, functional, and pharmacotherapeutic. The most common approach was physical therapy, usually operationalized by a physical therapist embedded in the MDC program. As an example of this focus on physical therapy, one program defined its core therapeutic goal to be "increase activity" and they used a structured approach prescribed by a physiotherapist "to increasing strength, range of motion and endurance" [85].

Another pain relieving mechanism was anesthetic interventional therapy. This was utilized prominently in the two sole studies which showed opioid dose reduction without any return to opioid use at 12 months following program completion [44, 101]. One study followed 103 Australian patients on long-term opioid therapy who received MDC together with a spinal cord stimulator (SCS) to help manage complex chronic pain [101]. Of these patients, 6 did not change their opioid doses, 53 discontinued their opioids, and 44 reduced their doses of opioids. A limitation of this study, however, was that patients with significant associated psychological symptoms, which is an important driver of opioid use [118–120], were excluded.

Acupuncture was another physical analgesic intervention which showed mixed effects on opioid dose reduction. One study examined the effect of auricular acupuncture and naloxone on rapid detoxification from opioids using tapering doses of methadone. Of 14 patients treated using this method, 12 were able to completely discontinue opioid use and all remained opioid abstinent at long-term follow-up between 6 and 15 months after the intervention [76]. A pragmatic trial assessing acupuncture in a community-based setting over a 6-day period demonstrated that 32% of patients self-reported a reduction in opioid use over the course of the trial [49]. Two studies of electroacupuncture, however, were not able to demonstrate any significant analgesic effect or any effect on opioid use [115, 116].

## Synthesis statement 2: Pain management using various modalities such as physical therapy, nerve blocks, and acupuncture is an essential component to MDC programs that reduce opioid doses. It was unclear which modality had the greatest opioid reducing effect in terms of number of respondents, total dose reductions or long-term abstinence

*Patient behavior change is universally present in MDC programs.* Most evaluations (66%) used a behavioural approach in their programs and 31 (30%) explicitly characterized the experience and care of chronic pain as being complicated by behavioural maladaptation. Chapman et al. [37] summarize this behavioural understanding by saying, "pain behaviors such as inactivity, verbal complaints, limping, taking drugs, and unemployment are behaviors that can persist because of environmental consequences." By facilitating change in behaviours from maladaptive to adaptive coping with chronic disease, programs improved quality of life [65, 77, 79]. Furthermore, given the influence of quality of life and psychological symptomatology on the subjective rating of pain severity, a behavioural approach can also improve pain severity [21, 30, 50].

This behavioural approach was often operationalized by having psychologists as members of the MDC team, which was found in 52% of the studies and occasionally as program directors [22]. Some programs go as far as training all team members in a behavioural approach to care, regardless of their professional background [38, 69, 70]. One evaluation noted, ""Mental health professionals within MPRPs [multi-professional rehabilitation programs] provide direct

clinical care but also guide the biopsychosocial model of pain management and cognitive behavioral interventions for multiple disciplines" [109].

Group and peer support, as well as family involvement, were additional behavioural components of many programs [21, 22, 28–30, 33–35, 37, 38, 40, 42, 44–48, 50–53, 56–61, 65–70, 73–75, 79–81, 83, 85, 87, 89, 90, 92–94, 96, 103, 104, 106, 108–112]. One program listed "agreement to family member or significant other involvement in treatment" as one of just five treatment inclusion criteria [61], another encouraged spouses to participate weekly in the inpatient program [22], and another included group psychotherapy for all patients [69, 70]. Group, peer, and family involvement is not a recent phenomenon as it was present in evaluations through each of the decades. This involvement was used primarily means for reinforcing behaviour change strategies. One representative program from 1979 that presages contemporary cognitive behavioural therapy used a three-stage psychoeducation approach to therapy in an outpatient setting [73]. This program focused on modifying psychological symptom control, cognitive interpretations of stimuli, and related behaviours. This program was successful in achieving significant dose reduction in most patients at program completion.

One study compared specifically the impact of a behavioural approach to a physical therapy only approach for the treatment of fibromyalgia [21] and found that only the behavioural group significantly changed its medication use. Likewise, they found more improvements in the behavioural group in terms of pain behaviours, sleep, and health care utilization.

## Synthesis statement 3: Patient-focused behaviour change is a very common intervention and frequently a guiding principle for MDC programs that result in opioid dose reduction

*Opioid dose reduction is not a passive "effect" of multi-modal interventions.* Besides aiming to improve pain outcomes and alter behaviours with respect to chronic pain, most programs (44 of 76) also actively intervened on opioid use, requiring patients to agree to a formal and pre-defined opioid taper in order to participate in the MDC program. In some cases, tapering was required as the first step of engagement in care, as opioid use was seen as an impediment to engagement with the rest of MDC care [37, 38, 69, 70]. The majority of programs (n = 41) were justified in terms of their impacts on opioid use which was often characterized as doing more harm than good for the treatment of chronic pain. As an example, Huffman [70] claims that "available evidence suggests that COT [chronic opioid therapy] does not provide long-term analgesia, improved function, and is associated with poor recovery." These kinds of claims were more common in the later literature, but were present even in some of the earliest literature included in this review. Writing in 1985, Kroening and Oleson [76] state in their justification for the program of rapid narcotic detoxification that, ". . .tolerance to the pain-relieving effects of the narcotic drugs and maladaptive behaviors related to drug dependence may negate the therapeutic value of taking these medications. Moreover, high doses of narcotic medications may interfere with the effectiveness of alternative methods of pain control."

There was a subset of articles in which there were MDC interventions that were successful in reducing pain or improving function, but did not have an integrated active opioid intervention and no resulting opioid dose reduction. One Danish study [71] conducted a long-term (10 year) follow-up of patients who had participated in an inpatient MDC program with a cognitive behavioural focus. The program did not include any specific opioid tapering component. In fact, ongoing opioid prescribing with a focus on using long-acting formulations was an important part of the treatment strategy [71]. At the 10-year mark, there was no significant change in opioid doses. In contrast, other medications that were commonly prescribed during the program, such as tricyclic antidepressants, had discontinuation rates as high as 76%. A

further three-armed Danish study compared a specialized chronic pain MDC program to a specialist-supported primary care MDC program to treatment as usual in primary care [32]. There was no opioid tapering component to any of these programs. The study found a 15/100 point improvement in visual analog scale for pain in the specialist based program as compared to no improvements in the supported primary care program or treatment as usual program. This is above the commonly used minimally important difference of 10/100 [8]. Likewise there were similar small improvements in overall psychological well-being in the specialist group and none in the other two groups. Despite these differences, there were no changes in opioid doses in any of the groups.

In one UK study [50], opioid tapering was not required for participation in the program, but was emphasized and recommended. Accordingly, the participants in this study had only modest reductions in opioid doses as compared to similar programs that required opioid tapering. This modest effect is echoed in another setting with a soft approach to tapering [84], where only 34.3% of patients showed any kind of opioid dose reduction. A further randomized control trial in the U.S. assessed the efficacy of optimizing antidepressant medication and providing chronic pain self-management education for improving depression in chronic pain [77]. Patients were recruited from and remained under the care of the primary care provider and antidepressant therapy was managed by a nurse under the supervision of a psychiatrist. The self-management program took a primarily behavioural approach to self-care and did not include any specific opioid management. While there were significant changes in antidepressant use and improvements in both depression and pain severity, there were no significant differences between the groups with respect to opioid use.

Furthermore, health system or even national orientation towards opioids for the management of chronic pain is a prevalent contextual factor. In included studies from countries such as Denmark [32, 63, 71], Germany [21, 78, 94] or the UK [19, 22, 31, 42, 50] where there were no prevailing concerns about opioid use, MDC programs did not employ, let alone require, opioid tapering. In such situations, there were no consistent reductions in opioid use, even when pain and behaviours improved. On the contrary, in American programs, where there are cultural and political concerns about opioid use, including in the form of significant national and regional opioid-related policies such as the declaration of a public health emergency by the Department of Health and Human Services [121], there were embedded and required opioid tapering programs which generally resulted in opioid dose reductions.

## Synthesis statement 4: Programs that do not directly manipulate opioid doses are unlikely to reduce opioid use. A greater focus on opioid dose reduction was associated with larger opioid dose reductions. National and health system attitudes towards opioids directly influence the presence and intensity of opioid tapering components

*There was self-selection in tapering programs.* Some programs, and therefore their evaluations, were based upon a participant pool who were already willing to taper, whether through voluntary actions or as a as a requirement of the program. Studies that included both patients on opioids as well as patients not on opioids, reported that higher proportions of non-completers were people on opioids. Likewise, other studies reported that those on higher doses of opioids or patients who were reluctant to change their opioids or existing pain management approach were less likely to complete the program [20, 47, 66, 70, 75, 87, 89, 108]. For example, Mehl-Madrona et al. [87] experienced some participant loss in their program from individuals who left to seek physicians who would prescribe opioids without restrictions. Additionally, of the studies that reported on non-completion (n = 41), some studies also reported discrepant

expectations of the program [20, 35, 43, 45, 48, 64–67, 108] and noncompliance [52, 63, 84, 89] as reasons for participant loss.

## Synthesis statement 5: MDC for opioid dose reduction has been studied with people who are willing to actively engage in tapering of opioids

*Treatment setting*. Most of the above described studies were in tertiary care settings to which patients were referred by their primary care providers. Thus, by definition, these programs involve a change in prescriber from primary care to tertiary care. It is unclear whether the referral process and the resulting change in prescriber is an important mechanism of change. Two studies set in primary care showed divergent but consistent results. One Canadian program offered a group-based behavioural program set in primary care [28]. The program was led by an experienced occupational therapist and social worker and required referral from the primary care physician who was also the primary opioid prescriber. While this program demonstrated some improvements in pain and number of clinic visits, the program did not have any effect on opioid use. Importantly, this program did not include a direct medication management component or involve a change in prescriber. Another study [95] examined a US Veterans Administration primary care based interprofessional pain treatment program delivered by a primary care physician with chronic pain expertise, a psychologist and a pharmacist, as well as expedited access to a recreational therapist. Patients had to be referred to the program by their primary care physician and opioid prescribing was taken over by the program physician for the duration of patients' participation. Upon program completion, the program physician also provided detailed long-term management recommendations for chronic pain management and opioid prescribing back to the referring primary care physician. This program was effective in reducing opioid doses. We identified no examples of programs that resulted in opioid dose reduction that did not involve a change in the opioid prescriber.

## Synthesis Statement 6: Treatment setting (tertiary versus primary care) is not determinative of outcomes. Programs that did not change the prescriber, regardless of treatment setting, were unlikely to be successful in reducing opioid doses

**Contexts: Health system resources and national attitudes influence program design.** It is evident from the findings described above that MDC in fact is not the mechanism through which change happens, but provides the context in which the mechanisms of pain relief, behaviour change, and provider/prescriber change can happen. The majority of the studies were in academic and tertiary care settings given the consolidation of chronic pain care, especially involving MDC, at this level of care. Such settings had the mandate, expertise, and resources to conduct the empirical investigations included in this review.

Likewise, we found some of the programs were of significant intensity. Of the 19 programs from which contact time could be extracted, 11 (58%) included 75 hours or more of contact time. All 11 of these programs were in tertiary care settings, and 9 were based at academic centres [21, 35, 38, 50, 70, 84, 89, 90, 111]. Only specialized centres are likely able to offer such time-intensive programs. Primary care, in contrast, is likely to be focused on more general and undifferentiated problems. The single primary care based program from which contact time could be extracted included 16 hours of contact. However, this program did not demonstrate any reductions in opioid use [28].

Both inpatient and outpatient care models had similar structures. One study specifically compared an inpatient to outpatient approach and found that while both showed significant improvements in medication use, physical performance, and psychological function,

inpatients made greater gains and were more likely to sustain these gains over the longer-term [22]. There are important differences between the approaches, in that inpatient programs provide a more controlled environment, but are most costly and there are concerns about translating improvements from inpatient settings back into normal life. Importantly, the inpatient program had four times the number of hours of programming as compared to the outpatient program. Therefore program intensity, or contact time, may be a more important contributor than specifically inpatient or outpatient delivery.

### Synthesis statement 7: Chronic pain MDC programs provide an amenable context for analgesia, patient behaviour change, and opioid tapering to co-occur. Given the contact time required for such programs, they are more likely to occur in resource intensive settings, such as tertiary care centres

**Outcomes: Opioid dose reductions do not always persist.**    Changes in opioid dose were dynamic over the periods of time studied. In many cases where there were significant opioid dose reductions at program completion, return to opioid use at 12 months was observed for some participants. Rates of return to opioid use varied, but patterned between about 22.5% [69] to as high as 41% [37]. Some studies attempted to determine the factors influencing the likelihood of return to use. These include pain severity at the time of opioid withdrawal [78], depression severity [69], and level of functional impairment [70]. None of the studies examined for pre-existing opioid use disorder as a factor.

Several studies have examined the issue of agency, and specifically self-efficacy, in determining longer-term outcomes. This is especially important when one of the primary approaches to change is a behaviour change approach that is meant to translate back into day-to-day life. One retrospective program evaluation found that self-efficacy was an important predictor of medication use at program completion and at follow-up (6–12 months post completion), though self-efficacy alone could not explain all the variance in the significant relapse rate [53]. Another study compared outcomes between groups who elected to reduce their opioid use by a classic "pain cocktail" (provider controlled) approach to a patient controlled approach [19]. They found that at program completion, opioid abstinence rates were higher in the provider-controlled group. Nevertheless, at 6 months post program completion, the abstinence rates in the two groups were equivalent and those who had been in the patient-controlled group tended to be on lower doses of opioids. It is important to note, however, since patients were allowed to select their method of dose reduction and the patient-controlled group did have significantly lower pre-treatment doses.

### Synthesis Statement 8: Return to opioid use after complete discontinuation was common. Factors that have been investigated as influencing the likelihood of return to use included pain levels, depression severity, functionality, and degree of self-efficacy

To summarize, the logic and design of MDC programs that reduce opioid use from which we can derive the primary mechanism is as seen in Fig 5.

## Discussion

This systematic rapid realist synthesis of MDC for opioid dose reduction has provided insights into the nature of these interventions and their utility for addressing contemporary opioid epidemics in the Global North. These lessons have been summarized in eight succinct synthesis statements. We now turn to the broader relevance of these findings.

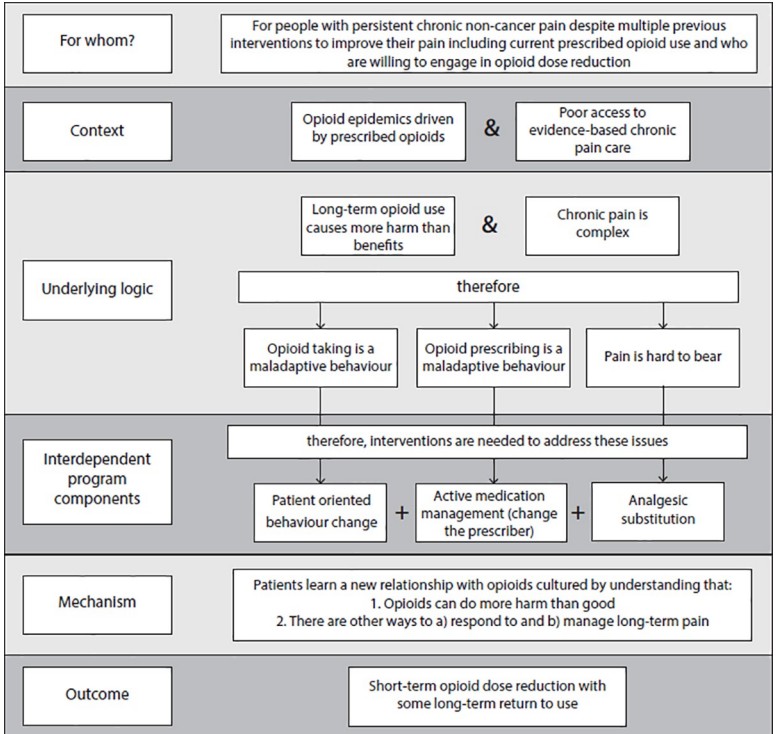

**Fig 5. Summary of the logic and design of MDC programs.**

First, the findings suggest that the analgesic substitution understanding of MDC is insufficient as an explanation for opioid dose reduction. Previous reviews of chronic pain MDC have also shown that these programs typically serve people who have lived with pain for many years [122]. For such patients who have been on chronic opioid therapy, providing alternate forms of analgesia (non-opioid pharmacotherapy or non-pharmacotherapy) is not sufficient for reducing opioid use. Other aspects of the program, namely a behavioural component and a direct opioid management component, are also required. We have emphasized this by understanding MDC not as the mechanism for change but instead as the context in which the collection of mechanisms can operate. By extension, then, improving access to quality chronic pain care may not, on its own, reduce opioid doses at a population level. This nuance is relevant for both practicing clinicians and health policy planners. While improving chronic pain care is important in its own right, this synthesis questions the direct impact this may have on current opioid epidemics. The temporal trend of the included studies is particularly telling here. There was a surge in publications on this topic in the early 1980s, a significant drop through the 1990s and then a resurgence in the 2000s and beyond. This has a specific relationship with the contemporary opioid crisis which very much ballooned in the early 2000s. Writing in 2013, Murphy et al. [89] state explicitly, "As concerns associated with sustained opioid analgesic use continue to build, alternative empirically supported approaches for treating or managing chronic pain should be considered more seriously. Interdisciplinary pain programs (IPPs) represent one of the best alternatives. IPPs have been found to improve functional status, reduce opioid analgesic medication use, improve psychologic well-being, and reduce pain severity." This suggests that the attention towards MDC for chronic pain, in the scientific literature at least, has returned more so as a means of responding to opioid epidemics, than because of its primary merit of improving chronic pain care.

Second, the collected literature embodies a persistent tension in the field as to the role of opioids in pain reduction and functional improvement [123, 124]. Writing in 2006, Maclaren et al. [84] note that "the use of opioids in the management of chronic non-cancer pain is controversial. Proponents suggest that pain relief can safely be achieved through the use of opioids, whereas opponents argue that opioid side effects and addiction potential may hinder their use." This difference of opinion is best exemplified by their starkly contrasting approaches to opioids between programs in Northern Europe and the US, but the similarity of the pain and function improvement components of their programs. Importantly, the specific orientation to opioids is truly determinative of the outcomes for opioid dose reduction. In the Northern European programs that do not directly manipulate opioid doses, there is no consequent opioid dose reduction, while the American programs typically require tapering as part of the participation in the program and there is consequent dramatic reduction in opioid doses.

Third, it is particularly noteworthy that essentially every program included in this review integrated a behavioural approach to care, frequently as the overriding philosophy of care. Most of this behavioural approach was directed towards improving maladaptive responses to chronic pain such as social isolation and avoidance of physical activity. Particularly in the older literature included in this review from the 1970s and 1980s, program and program planners also conceptualized medication-taking as a maladaptive behavior. This conceptualization has deep roots in the MDC for chronic pain literature, including some of the foundational studies that were included in this review [58]. By conceptualizing medication use not just as a means of analgesia but as a learned, maladaptive behavioural response to medicalized chronic pain, opioid use and opioid dose reduction come under the purview of behaviour change strategies [37, 38, 85]. This behavioural approach to medication use has been noted for other kinds of medication besides opioids. As medication utilization and appropriateness generally, i.e. outside of opioids, has risen to the top of health system agendas internationally, this finding emphasizes that behavioural approaches could play an important system-level role. Indeed other studies outside of opioids have identified the important role of addressing modifiable behaviours in addressing medication use and adherence [125].

While the findings of this review suggest that active opioid tapering programs, i.e. provider-centred behaviours, are essential in reducing opioid doses, these findings also suggest that a behavioural approach to improving medication-taking offers an important opportunity for patient-centred care and supporting increased self-efficacy [126]. The included studies did not typically follow the patients over longer term, i.e. after they left the controlled environment of the program, and there was therefore insufficient investigation of factors that influence long-term opioid dose reduction. Since the primary impetus of opioid dose reduction is long-term risk reduction, there is a persistent knowledge gap of factors besides providing a time-limited controlled environment for change that are important for achieving long-term changes in opioid use and thus opioid-related harms. Self-efficacy may be an important mechanism for achieving this kind of long-term change [104] as has been demonstrated in comparable populations such as those living with chronic pain, multiple chronic diseases, polypharmacy, substance dependence and those aiming to create positive lifestyle changes [127–131].

In the specific temporal and geographic contexts of the studies where there was limited access to non-prescribed high dose opioids, the rate of return to opioid use of 20–40% could be an acceptable outcome. This is particularly so since as the return was usually to a lower dose than at program initiation. However, in the contemporary landscape such a rate of return to opioid use could be catastrophic and could undo the relatively small potential population level benefits of opioid dose reduction [132]. Currently in jurisdictions throughout North America there is diminishing access to prescribed opioids, i.e. opioids that are regulated and have known potencies, and higher access to illicit street opioids, i.e. substances that are not

regulated and whose potencies are uncertain [4]. If patients were to access the contemporary illicit street supply of opioids after completing a taper they would be at a very high risk of overdose. Before embarking on wide-spread implementation of MDC for opioid dose reduction, then, it is imperative to develop a greater understanding of the factors predicting return to use, so that protections can be built into such programs.

Consideration must be given to the settings in which MDC programs for opioid dose reduction can and should be deployed. The majority of long-term opioid prescribing is done in primary care. Furthermore, there are initiatives underway in many jurisdictions to improve primary care level access to MDC for chronic diseases, including chronic pain [133–136]. Likewise, many jurisdictions, including in better resourced health systems in the Global North, have persistently poor access to specialized, multidisciplinary chronic pain care [137]. Expansion of access to such programs will be a resource- and time-intensive process. Thus, it is natural to think that primary care level MDC would be the ideal setting to achieve opioid dose reduction, and some initiatives have already been made in this direction [25]. However, the results from this review are quite clear that such programs will not be effective in reducing opioid use unless there is a change in the primary opioid prescriber and that this provider should have some additional expertise in chronic pain and opioid prescribing. As such, MDC settings in primary care should provide a mechanism for changing the prescriber with more expertise, even temporarily during active involvement in a program, to achieve the intended results. This could be operationalized in many ways, such as modified shared-care models which have been widely deployed to improve primary care access and capacity for mental health in Canada [138]. Likewise, persistent imbalances in institutional support (such as funding) for tertiary versus primary care would need to be acknowledged and addressed to develop and support any such programs over the long-term. Though no examples of this were found in this review, it is also possible that providing MDC for patients together with interventions (educational, behavioural, etc.) that target behavior change in the primary prescriber may be able to achieve the intended outcome of long-term opioid dose reduction. All such modified strategies would require extensive evaluation prior to wide dissemination and this review provides some important principles for guiding such evaluations. Likewise, any strategies need to accommodate regional and national differences between health systems. For example, compared the United States, Canadian medical care is distinguished by a single public-payer system and a greater gatekeeper function of primary care physicians.

## Strengths and limitations

The primary strength of this review is that we have synthesized evidence from a heterogeneous set of evaluative literature. We have identified program regularities, and examined several counterfactuals, related to opioid dose reduction from literature spanning nearly half a century, 9 countries, and multiple study designs. Many of the reviewed programs evolved over time, and under real world conditions. Given the purpose of this systematic rapid realist review—to use interpretive frameworks leading to an understanding of what works, for whom and under what conditions—a diverse literature pool was purposefully included. In contrast to a traditional systematic review focused on aggregating magnitudes of effects, we did not conduct a quality appraisal of the included literature. This type of assessment would not have been consistent with accepted methods, and more importantly would likely have excluded materials that proved essential in understanding factors such as contexts and mechanisms [139].

To our knowledge, this is the first study to synthesize data on multidisciplinary care for opioid dose reduction beyond program description. Likewise, we have contextualized the significance of this synthesis for contemporary health care program design and evaluation within the complex and shifting dynamics of global opioid epidemics. This was facilitated by involving a

research team, local reference panel, and international experts with highly relevant but diverse expertise in the areas of lived experience with opioid dose reduction, chronic pain management, opioid and mental health policy, multidisciplinary care, primary care, and guideline development.

There are a few important limitations relating to our search strategy. First, due to resource limitations, we examined only the English language literature. This certainly would have influenced the national origin of the included studies and likely increased the number of included articles from the US. Given the variable national perspectives on the role of opioids seen even within this sample, particularly the contrast between Northern European and American programs, it is plausible that including non-English studies would have included different kinds of approaches to opioid dose reduction and multidisciplinary care, and thus affected theory development. Second, while our formal database search used five independent bibliographic databases, we did not screen results from an Ovid EMBASE search, besides the conference and meetings abstracts. We felt that there was sufficient duplication with results from existing databases. Nevertheless, it is possible that important program evaluations were missed by excluding the search. Given the strong consistency within the included studies, however, we are confident that additional records from a wider search would not have substantially changed theory development. Third, our search strategy was dependent on the adequate construction of two very complex search concepts, namely those of multidisciplinary care and dose reduction. There were no extant relevant search concepts to draw from, so we instead undertook extensive consultation and validated our search against a large set of relevant articles from a recent systematic review and a recent clinical practice guideline. Despite this extensive process, our findings could in part be artefactual of definitions of these two key concepts.

Our process of generating theoretical understandings was interpretative and inductive and aimed to stay within the specific interpretation of realism as espoused by Pawson and Tilley. Nevertheless, it is possible that another team of researchers could have derived another program logic and CMO configuration from this same data set. The particular challenge of defining mechanisms, and sorting between mechanisms and contexts has been noted by others [140]. Further to this, we noted, specifically, that different mechanisms could be derived from these program evaluations depending on the research team's view of what is most real in realism. Dalkin [15] notes varieties of interpretations of the locus of causal mechanisms, from the structural component of the social world of Bhaskin to the cognitive locus of Pawson and Tilley. The scope of reality, however, may shift dramatically as realist approaches become increasingly applied beyond social programs to medical programs, which inhabit biological and physical worlds as well as social worlds. For this review, this issue is most evident in the interpretation of the analgesic substitution program component. We have aimed to capture this as a cognitive mechanism, namely that providing alternative methods of pain relief teaches a person with pain that opioid use is not necessary. However, a physical or biological realist may easily argue that the mechanism may be better captured by understanding how an anaesthetic agent (e.g. from a nerve block) or electrical current (e.g. from a spinal cord stimulator) facilitates changes at the level of the central and peripheral nervous systems which replace or obviate the need for opioid receptor stimulation by oral opioid analgesic medications. The choice of mechanism, then, appears contingent on one's interpretations of realism, the possibility of strata of reality and how these strata relate in dependent or interdependent ways. This suggests that as realist methods become increasingly applied in medical settings, researchers must be more explicit about the theory of reality and the concomitant theory of causation and mechanism that they will employ. We aim to have addressed this limitation primarily by conducting a rapid realist review which focuses less on theory development and defining specific mechanistic understandings. We believe the review findings more focused on a specific outcome and

associated program components are widely relevant despite this apparent pluripotency of kinds of mechanisms at play.

## Conclusion

This review has distilled three interdependent program components found in multidisciplinary care programs that reduce opioids doses: pain and functional improvement, patient behavior change, and changing the opioid prescriber. This distillation can be used to inform and evaluate health system responses to opioid epidemics in the Global North and elsewhere. This review also identifies and highlights important, long-standing and ongoing tensions between chronic pain and opioids in the spheres of clinical care, health systems and health policy. Such tensions directly impact on the design, delivery and subsequent outcomes of multidisciplinary care programs for people living with chronic pain who use opioids long-term, and thus must be explicitly addressed by program and policy developers.

## Supporting information

**S1 Table. OVID Medline search strategy.**
(DOCX)

**S2 Table. PRISMA checklist.**
(DOCX)

## Acknowledgments

AS would like to thank Mr. Yarema Gribowski for input into previous drafts of this manuscript. The authors would also like to acknowledge Ms. Mary Nelson, Dr. Bryan MacLeod, Dr. Kim Corace, Dr. Dino Smiljic, Dr. Frank Joseph and members of the Canadian National Pain Faculty for providing invaluable expert consultation at various stages of the review process.

## Author Contributions

**Conceptualization:** Abhimanyu Sud, Shawn Tracy, Ross Upshur, Michelle L. A. Nelson.

**Data curation:** Alana Armas.

**Formal analysis:** Abhimanyu Sud, Alana Armas, Heather Cunningham, Shawn Tracy, Kirk Foat, Navindra Persaud, Fardous Hosseiny, Sylvia Hyland, Leyna Lowe, Erin Zlahtic, Rhea Murti, Hannah Derue, Ilana Birnbaum, Katija Bonin, Ross Upshur, Michelle L. A. Nelson.

**Funding acquisition:** Abhimanyu Sud, Heather Cunningham, Shawn Tracy, Kirk Foat, Navindra Persaud, Fardous Hosseiny, Sylvia Hyland, Ross Upshur, Michelle L. A. Nelson.

**Investigation:** Abhimanyu Sud, Alana Armas, Shawn Tracy, Erin Zlahtic, Rhea Murti, Hannah Derue, Ilana Birnbaum, Katija Bonin, Michelle L. A. Nelson.

**Methodology:** Abhimanyu Sud, Heather Cunningham, Ross Upshur, Michelle L. A. Nelson.

**Project administration:** Alana Armas.

**Resources:** Ross Upshur.

**Supervision:** Abhimanyu Sud, Ross Upshur, Michelle L. A. Nelson.

**Validation:** Heather Cunningham, Kirk Foat, Navindra Persaud, Fardous Hosseiny, Sylvia Hyland, Leyna Lowe.

**Visualization:** Abhimanyu Sud, Alana Armas, Erin Zlahtic, Rhea Murti, Hannah Derue, Ilana Birnbaum, Katija Bonin, Michelle L. A. Nelson.

**Writing – original draft:** Abhimanyu Sud, Alana Armas, Ross Upshur, Michelle L. A. Nelson.

**Writing – review & editing:** Heather Cunningham, Kirk Foat, Navindra Persaud, Fardous Hosseiny, Sylvia Hyland, Leyna Lowe, Erin Zlahtic, Rhea Murti, Hannah Derue, Ilana Birnbaum, Katija Bonin.

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
