## [Decision Letter · Decision Letter 0]

14 Apr 2020

PONE-D-20-01387

Multidisciplinary care for opioid dose reduction in patients with chronic non-cancer pain: A systematic realist review

PLOS ONE

Dear Dr. Sud,

Thank you for submitting your manuscript to PLOS ONE. After careful consideration, we feel that it has merit but does not fully meet PLOS ONE’s publication criteria as it currently stands. Therefore, we invite you to submit a revised version of the manuscript that addresses the points raised during the review process.

We would appreciate receiving your revised manuscript by May 29 2020 11:59PM. To enhance the reproducibility of your results, we recommend that if applicable you deposit your laboratory protocols in protocols.io, where a protocol can be assigned its own identifier (DOI) such that it can be cited independently in the future. For instructions see: http://journals.plos.org/plosone/s/submission-guidelines#loc-laboratory-protocols

We look forward to receiving your revised manuscript.

Kind regards,

Vijayaprakash Suppiah, PhD

Academic Editor

PLOS ONE

Journal Requirements:

2. Please report the date of last search, ensuring that it has been performed in order to allow the inclusion of studies published in the past 12 months. In addition, please discuss the quality and biases introduced by each of the included studies, or justify why this has not been performed.

Reviewers' comments:

Reviewer's Responses to Questions

**Comments to the Author**

1. Is the manuscript technically sound, and do the data support the conclusions?

Reviewer #1: Yes

Reviewer #2: Partly

Reviewer #3: Yes

2. Has the statistical analysis been performed appropriately and rigorously? 

Reviewer #1: Yes

Reviewer #2: N/A

Reviewer #3: N/A

3. Have the authors made all data underlying the findings in their manuscript fully available?

Reviewer #1: Yes

Reviewer #2: Yes

Reviewer #3: Yes

4. Is the manuscript presented in an intelligible fashion and written in standard English?

Reviewer #1: Yes

Reviewer #2: Yes

Reviewer #3: Yes

5. Review Comments to the Author

Reviewer #1: A few comments which need attention:

Line 51-52, "Unintentional drug overdose deaths are high and on the rise throughout the Global North. The United..."  Please define Global North for this article

Line 55-56, "One large socioeconomic effect of these epidemics is that life expectancies in the US and Canada have dropped over the last several years Please re-word to remove "dropped", perhaps say "decreased"

Line 78-79 "for tapering, or dose reduction. The Canadian guidelines provide a recommendation based on weak evidence  Please better define or elaborate on "weak"

Line 139-140 "Grey literature was searched from a variety of sources including Ovid EMBASE (conference proceedings and meeting abstracts), Define "grey"

Line 183-184 "A data extraction form was created in Excel and tested by multiple reviewers. Once validated, the form was replicated in Covidence and data extraction was conducted singly. How many reviewers? how was it validated?

Line 367-368,"On the contrary, in American programs, where there are cultural and political concerns about opioid use, there were" please elaborate on "political concerns"

Line 476-477, "Interdisciplinary pain programs (IPPs) represent one of the best alternatives."

 Throughout this paper the focus is on multidisciplinary, rather than interdisciplinary, approach to pain and opioid reduction. These are two different models and approaches to pain care, both with pros and cons. This article should specifically define these two, and explain why multidisciplinary versus interdisciplinary was chosen as the methodological approach.

Lines 543-546 "This could be operationalized in many ways, such as modified shared-care models which have been widely deployed to improve primary care access and capacity for mental health in Canada [134]. Likewise, persistent imbalances in institutional support (such as funding) 

The authors should add some discussion about (1)payment reimbursement models, single payer systems versus private insurance payers from region to region, and how this might impact opioid tapering as well as patient and provider resources, treatments, incentives, and patient outcomes. In the US, there is a lack of incentive in this regard, thus discussing this for the US and other populations would be important as well as (2) lack of pain related education in medical schools and residencies which could also impact opioid tapering in various populations

Reviewer #2: Thank you for the opportunity to review the manuscript entitled "Multidisciplinary care for opioid dose reduction in patients with chronic non-cancer pain: A systematic realist review". Overall, the manuscript is very well-written. It is also generally easy to follow the authors’ logic.

• A major point of constructive criticism lies with one of the main tenets of the review: Specifically, that effective programs (those that results in sustained reduction of opioid use), necessitate a “change in the opioid prescriber.” Although the “realist” portion of the review is understandably fraught with challenges for those who rely upon rigorous statistical analyses to generate hypotheses, this seems like an incorrect and far too simplistic conclusion.

• Based on the authors’ summaries, this seems much more like an issue of expertise, rather than a simple change in opioid prescriber/provider.

• The statement that “…results from this review are quite clear that such programs will not be effective in reducing opioid use unless there is a change in the primary opioid prescriber” (starting at line 541) seems unfounded.

• The authors more/less acknowledge that their conclusion may be oversimplified. For example by stating that the change in opioid use may be explained by a different mechanism, such as the referral process itself (see lines 387-388). Nonetheless, the authors go on to explicitly say that primary care should input a mechanism for changing providers, as if what was found in the review was a collective result of a simplistic, literal change in the person prescribing the opioid.

• A change in providers is accompanied by myriad other changes, including the potential for a change in the patient’s psyche as a result of the whole referral process, the patient’s orientation toward “getting well,” etc.

• The most important change that accompanies a change in provider however, appears to be a vast increase in the expertise of the providers. From this reviewer’s perspective, that should be the primary point, rather than a simple change in providers.

Minor points:

• It would be helpful to have a table with all 8 synthesis statements.

• The right side of Table 1 is cut off.

Reviewer #3: This is an excellent review and contribution to the literature of opioid use in patients with chronic pain.

I have a few minor questions and comments:

Page 2 line 63 reference? Please place reference here

Page 3 line 73-74, this sentence is unnecessarily accusatory. The opioid crisis is complicated it is not entirely iatrogenic.

Page 3, line 83-4, sentence is poorly written Page 3 line 94: do not capitalize the W in What

Table 1 did not fit on the page 8-13

Page w13: not sure that a program that ran for 1-5 days is in any way similar to longer running program. Please explain why this was included:

Page 14: Lines 229and 230. Confusing sentence

Page 15: Participants: We would expect different outcomes for patients with any form of opioid use disorder compared to those who are on chronic opioids for chronic pain. Please explasin why it is acceptable to lump them all together for the purposes of this study.

Page 17, line 299: Please define MPRP

Page 18, line 327: define COT-

Page 20-1, lines 376-378, “Some studies did report that higher proportions of non-completers were people on opioids…..” this is confusing, since the purported purpose of the this paper is to investigate the success of tapering opioids, why would the authors be looking at studies of patients who are not on opioids?

Page 21, line 381 clarify opiates versus opioids

Page 23, lines 434-436 Did any of the programs look at preexisting opioid use disorder?

Page 29-30. Excessively and unnecessarily long,

6. PLOS authors have the option to publish the peer review history of their article (what does this mean?). If published, this will include your full peer review and any attached files.

Reviewer #1: No

Reviewer #2: No

Reviewer #3: Yes: Rita Agarwal

---

## [Author Response · Author response to Decision Letter 0]

13 May 2020

Thank you for the excellent review comments which have helped to improve the manuscript. We have submitted our responses to the review comments as a separate document.

---

## [Decision Letter · Decision Letter 1]

29 Jun 2020

PONE-D-20-01387R1

Multidisciplinary care for opioid dose reduction in patients with chronic non-cancer pain: A systematic realist review

PLOS ONE

Dear Dr. Sud,

Thank you for submitting your manuscript to PLOS ONE. After careful consideration, we feel that it has merit but does not fully meet PLOS ONE’s publication criteria as it currently stands. Therefore, we invite you to submit a revised version of the manuscript that addresses the points raised during the review process.

Please address the comments of the second reviewer. 

Thank you. 

We look forward to receiving your revised manuscript.

Kind regards,

Vijayaprakash Suppiah, PhD

Academic Editor

PLOS ONE

Reviewers' comments:

Reviewer's Responses to Questions

**Comments to the Author**

1. If the authors have adequately addressed your comments raised in a previous round of review and you feel that this manuscript is now acceptable for publication, you may indicate that here to bypass the “Comments to the Author” section, enter your conflict of interest statement in the “Confidential to Editor” section, and submit your "Accept" recommendation.

Reviewer #1: All comments have been addressed

Reviewer #2: (No Response)

2. Is the manuscript technically sound, and do the data support the conclusions?

Reviewer #1: Yes

Reviewer #2: Partly

3. Has the statistical analysis been performed appropriately and rigorously? 

Reviewer #1: Yes

Reviewer #2: N/A

4. Have the authors made all data underlying the findings in their manuscript fully available?

Reviewer #1: Yes

Reviewer #2: Yes

5. Is the manuscript presented in an intelligible fashion and written in standard English?

Reviewer #1: Yes

Reviewer #2: Yes

6. Review Comments to the Author

Reviewer #1: The authors did a thorough and detailed review of my comments and did an excellent job addressing my feedback and updating their manuscript. It is now a thoughtful and well constructed piece which will add nicely the literature regarding clinical opioid management.

Reviewer #2: Overall, the authors did a good job of addressing reviewers' comments.

However, I continue to think that Summary Statement #5 is overly simplistic. In the example included on page 23, while the occupational therapist and social worker may have been experienced, neither was a pain physician. In the second example on page 23, it is unclear when a “different provider” (from page 23: “…however, the opioid prescribing was taken over by the different family physician…”) took over. Together, these two examples do not convince the reader that simplistically changing a provider would significantly reduce opioid consumption.

Perhaps clarifying or using different examples would help make the point?

7. PLOS authors have the option to publish the peer review history of their article (what does this mean?). If published, this will include your full peer review and any attached files.

Reviewer #1: No

Reviewer #2: No

---

## [Author Response · Author response to Decision Letter 1]

3 Jul 2020

Thank you for the review and for identifying this concern.

We have clarified in the narrative preceding Summary Statement #6 that though the outcome between the two examples given was divergent (Seal 2017 resulted in opioid dose reduction and Angeles 2013 did not), these outcomes are both consistent with the need to change the prescriber to achieve reductions in opioid doses.

For the second example (ref 95, Seal 2017), we have clarified that the primary care physician with special expertise in chronic pain took over opioid prescribing at the time of referral to the Integrated Pain Team and also provided detailed recommendations back to the referring family physicians for long-term management. We hope this helps to clarify this process.

We have clarified in the first example (ref 28: Angeles 2013) that program participation required referral from the primary care physician. This program included an experienced occupational therapist and social worker (so was multidisciplinary, but without a physician), required referral by the primary care physician, resulted in some improvements in pain and health utilization, but did not result in any changes in opioid doses. Thus, we identified this is a good counterfactual example suggesting that without changing the prescriber (as was done in the example above, Seal 2017), opioid doses are unlikely to change.

As noted in our previous response and in the manuscript, since the vast majority of evaluations were based in tertiary care settings without reference to the primary care provider / primary opioid prescriber or referral process, there are no further examples for us to draw on to clarify this synthesis statement. However, we can have confidence in this statement since the majority of programs (as identified in synthesis statement #4) did involve active medication management, i.e. taking over prescribing from the previous prescriber during the patient’s participation in the program. In addition, we did not identify any contradicting examples.

It is important to note that our identified key program components (behaviour change, active medication management / changing the prescriber, and analgesic substitution) are described as interdependent - each are posited as necessary but none are independently sufficient in driving changes in opioid doses. We are not claiming that simply changing a provider would reduce opioid consumption, but that changing the prescriber is one essential component for multidisciplinary care programs to result in opioid dose reductions. If changing the pain provider was a necessary and sufficient condition, then we would expect all multidisciplinary care programs to result in opioid dose reductions – which is certainly not consistent with the included literature. Further empirical work making use of these findings will be essential in refining the set of necessary conditions and their relations with one another.

---

## [Editor Report · Decision Letter 2]

8 Jul 2020

Multidisciplinary care for opioid dose reduction in patients with chronic non-cancer pain: A systematic realist review

PONE-D-20-01387R2

Dear Dr. Sud,

We’re pleased to inform you that your manuscript has been judged scientifically suitable for publication and will be formally accepted for publication once it meets all outstanding technical requirements.

Kind regards,

Vijayaprakash Suppiah, PhD

Academic Editor

PLOS ONE

---

## [Editor Report · Acceptance letter]

10 Jul 2020

PONE-D-20-01387R2 

Multidisciplinary care for opioid dose reduction in patients with chronic non-cancer pain: A systematic realist review 

Dear Dr. Sud:

I'm pleased to inform you that your manuscript has been deemed suitable for publication in PLOS ONE. Congratulations! Your manuscript is now with our production department. 

Kind regards, 

on behalf of

Dr. Vijayaprakash Suppiah 

Academic Editor

PLOS ONE